# The Stem Cell Revolution Revealing Protozoan Parasites’ Secrets and Paving the Way towards Vaccine Development

**DOI:** 10.3390/vaccines9020105

**Published:** 2021-01-31

**Authors:** Alena Pance

**Affiliations:** The Wellcome Sanger Institute, Genome Campus, Hinxton Cambridgeshire CB10 1SA, UK; alena.pance@sanger.ac.uk

**Keywords:** protozoan parasites, stem cells, induced pluripotent stem cells, organoids, vaccines, treatments

## Abstract

Protozoan infections are leading causes of morbidity and mortality in humans and some of the most important neglected diseases in the world. Despite relentless efforts devoted to vaccine and drug development, adequate tools to treat and prevent most of these diseases are still lacking. One of the greatest hurdles is the lack of understanding of host–parasite interactions. This gap in our knowledge comes from the fact that these parasites have complex life cycles, during which they infect a variety of specific cell types that are difficult to access or model in vitro. Even in those cases when host cells are readily available, these are generally terminally differentiated and difficult or impossible to manipulate genetically, which prevents assessing the role of human factors in these diseases. The advent of stem cell technology has opened exciting new possibilities to advance our knowledge in this field. The capacity to culture Embryonic Stem Cells, derive Induced Pluripotent Stem Cells from people and the development of protocols for differentiation into an ever-increasing variety of cell types and organoids, together with advances in genome editing, represent a huge resource to finally crack the mysteries protozoan parasites hold and unveil novel targets for prevention and treatment.

## 1. Introduction

Millions of people worldwide are affected by infectious diseases caused by protozoan parasites, which have been a major public health problem, particularly in underdeveloped countries. The problem is exacerbated by the scarce availability of vaccines and drugs to treat these diseases, many of which have poor efficacy and secondary effects of varied severity. The appearance and spread of drug resistance and the lack of targets for vaccine development make the search for solutions very difficult.

In order to find target genes and proteins for the development of novel methods of intervention, it is fundamental to understand the biology of these organisms and specifically, the mechanisms of their interaction with the human host. One of the essential tools for this purpose is the ability to maintain these parasites in culture and reproduce their behaviour in the laboratory. A number of cellular culture systems have been used over the last decades that allow controlled, detailed observation of the parasites and the possibility of genetic modification, which improved dramatically with the development of CRISPR/Cas9 technology. These tools have facilitated the identification of novel parasite genes and their function [1]. However, culturing parasites is not always easy or even possible. For example, *Cyclospora cayetanensis* [2] or the malaria species *Plasmodium vivax* [3] cannot be cultured in vitro at all, while for other parasites such as *Cryptosporidium* spp [4], *Giardia duodenalis* [5], *Trypanosoma cruzi* [6], among others, existing in vitro or in vivo culture systems are highly unsatisfactory [7]. Either because parasites are cultured in cells they do not naturally infect or because immortalised or cancer cell lines are used that have diverted from the original cell type or because one cell type in a monolayer does not reproduce the natural environment of the parasite, most culture methods remain highly artificial. This limitation is further increased by the complex life cycle of protozoan parasites, often involving tissues difficult to obtain or replicate in vitro.

While animal models have provided a wealth of information about diseases such as malaria, they rely on species-specific parasites, such as the murine *P. berghei*, *P. yoelii* and *P. chabaudi* in the case of malaria, which are very different from the parasites infecting humans and, therefore, the knowledge these models provide is not necessarily transferable to the human scenario [8]. Furthermore, animal models are expensive and labour-intensive, more so for some parasites than others, such as the use of neonatal calves [9] and gnotobiotic piglets [10] to study the pathology of human cryptosporidiosis.

The infections by protozoan parasites can have a wide range of clinical manifestations ranging from acutely lethal, to chronic, debilitating or asymptomatic, which impact the prevalence of these diseases and also the research into understanding them. The expansion and improvement of whole genome sequencing and its use to develop genome-wide association studies (GWAS) have changed the way human disease is studied [11]. The association of certain genotypes to resistance or susceptibility to specific diseases is a powerful tool to discover human factors involved in the infection. Existing cellular systems and animal models may not give an insight into the impact of the human genomic background and its variability on disease.

The propagation of pluripotent cells from human blastocysts that could be cultured on mouse embryonic fibroblasts (MEF) [12] led to a protocol for the derivation of human embryonic stem cell (ESC) lines that have the capacity to grow indefinitely and differentiate into any cell type. This breakthrough was followed by the discovery that terminally differentiated cells can be reprogrammed back to pluripotency [13] by transduction with specific transcription factors: Oct-4, Sox-2, KLF-4 and c-myc, to generate induced pluripotent stem (iPS) cells. This novel technology opened a whole new approach for the study of human disease, including protozoan infection. With this system, it becomes possible to generate inaccessible cell types in vitro that can be used to culture parasites in their right environment, these cells can be genetically modified to study the host genes involved in the disease directly and iPS cells can be derived from people with specific genetic characteristics and particular variants of certain genes to start unravelling how the genetic background influences the development of the disease. Furthermore, these cells can be stored for further use, thereby preserving the genetic characteristics in the studies performed. A further crucial development, particularly for the study of invasive parasites, was the discovery that stem cells can organise themselves into complex structures with all the cell types and characteristics of an organ. These organoids can be generated from ESCs, iPS [14,15] as well as from primary tissue [16], offering for the first time the possibility of reconstructing the composition and environment of an organ in vitro.

This review summarises the application of stem cell technology for the understanding of protozoan infectious diseases so far. The aim is to highlight how the new knowledge acquired with these approaches is helping in the understanding of parasite–host interactions and identification of novel targets for therapies and vaccines. These novel models are predicted to be instrumental in the development of new strategies for treatment and prevention of these worldwide diseases.

## 2. Specific Cell Type Generation: Stem Cells

Embryonic Stem Cells (ESC) are derived from the inner cell mass of the blastocyst at the early stages of embryogenesis. As such they are diploid, pluripotent and self-renewing, which means that they can be stored, cultured and expanded indefinitely maintaining these properties, making genetic manipulation possible and preservable. Their pluripotency gives them the ability to give rise to all three germ layers: ectoderm, mesoderm and endoderm, from which potentially all cell types can be derived [17] (Figure 1). Human ESCs pose ethical issues which limit their use to cell lines already derived and available under strict ethical approval. Less ethically challenging is the use of adult stem cells such as Haematopoietic Stem cells (HSCs), but these are much more difficult to isolate and cannot be maintained in culture for indefinite periods of time [18].

The ethical issues of embryonic stem cells have been avoided with the reprogramming of terminally differentiated cells back into pluripotency to generate iPS cells [13]. The reprogramming technology has been improved and expanded from the original fibroblast source to blood, which is much easier and less invasive to obtain [19]. iPS cells can also generate the three germlines and have the potential to differentiate into any desired cell type, with differentiation protocols constantly being developed and improved [20]. This technology also provides a unique opportunity to examine the effects of human genetic variation on infectious organisms and diseases and confirm correlations from epidemiologic and Genome-Wide Association Studies (GWAS).

### 2.1. Malaria

Despite substantial progress in recent years in controlling this disease, malaria is still the major protozoan infectious health burden affecting the world, leading to around half a million deaths every year [21]. Malaria is transmitted by infected female mosquitoes that deposit parasites in the skin during blood meals. From the skin, the parasite migrates to the liver, multiplies in the hepatocytes and enters the bloodstream as merozoites that infect erythrocytes, initiating the blood cycle. Human malaria is caused by 5 species of *Plasmodium* parasites of which *P. falciparum* is the most common and deadly while *P. vivax* and *P. ovale* form hypnozoites or dormant hepatic forms that cause recurrences of the disease and are difficult to clear. Though a vaccine (RTS,s/AS01) has recently been licensed to be deployed among African children [22], it has shown an efficacy of 30–50% so far, which is attributed to the immense genetic variability of worldwide populations of parasites [23]. The drugs currently used are losing efficacy due to increasing resistance of parasite strains, which makes new approaches an urgent public health priority. Of the 5 species, only *P. falciparum* and *P. knowlesi* can be cultured in vitro and only the blood cycle can be reproduced, while the whole life cycle can be studied using mouse models and more recently humanized mice [24,25] and non-human primate [26] models have been developed for human malaria parasites. However, the murine *Plasmodium* species behave differently from the human ones and many aspects of the disease in animal models are not representative of human physiology. Furthermore, though both *P. falciparum* and *P. knowlesi* have been readily genetically modified [27,28], editing of the host erythrocytes is impossible due to their enucleated nature, which makes an assessment of the impact of host factors on the disease very difficult to study.

Malaria is perhaps the infectious disease with the widest application of stem cell technology to answer some of the outstanding questions. An initial effort to manipulate gene expression in the anuclear erythrocytes used HSCs and RNAi-based knockdown strategies to screen for host determinants of malaria infection. An shRNA library targeting the erythrocyte-specific blood groups was transduced into CD34^+^ HSCs, which were then induced to proliferate and differentiate towards the erythroblast stage. Infection with fluorescent *P. falciparum* parasites and quantification of each shRNA abundance in the infected population of cells identified CD55 as an important host factor for *P. falciparum* invasion involved in parasite attachment to the erythrocyte [29]. This approach was also used to reduce the levels of Glycohporin A (GYPA) and demonstrate the importance of this erythrocytic receptor for parasite invasion [30]. It also provided crucial support for the finding that Basigin (BSG) is the receptor for *P. falciparum’s* Rh5 and is the universal essential protein for the invasion process [31]. These findings are at the basis of a novel vaccine formulation based on Rh5 [32,33], which is much more conserved among parasite strains around the world than other widely tested parasite proteins [34]. Further studies, performed with an immortalised adult erythroblast cell line [35], used CRISPR-mediated editing to generate BSG truncated mutants to study the mechanisms of this receptor [36]. They showed that the cytoplasmic domain of BSG does not participate in invasion and ruled out a role for cyclophilin B, suspected to participate as an interacting partner of BSG. The *P. falciparum* ligand of BSG, Rh5 has shown induction of antibodies with cross-strain in vitro growth inhibitory activity in a phase I clinical trial [33], which has generated great interest and hope for the future. Immortalised HSCs generated from peripheral blood were able to support invasion by *P. falciparum* and also *P. vivax* [37]. Knock out of the main erythrocytic receptor for *P. vivax*, the Duffy Antigen Receptor for Chemokines (DARC), in these cells strongly inhibited invasion which was restored by complementation, proving the approach in this species [37]. We have developed a general protocol to differentiate hESC as well as hiPS cells towards the haematopoietic path to generate erythroid cells able to support *P. falciparum* invasion [38]. Our studies demonstrate the feasibility of using this in vitro model and show that patient-derived iPS cells can be useful to establish the mechanisms of some well-known human variants, such as deletion of the Haemoglobin-α gene.

The liver stage of malaria parasites is very important, not just as the initiating point of infection but also because there are fewer parasites that are more exposed to the immune system at this stage and blocking it would efficiently avoid all phases of the disease, i.e., infection, clinical symptoms and transmission. Understanding the mechanisms and proteins involved in this stage is fundamental to identify vaccine and drug targets, including those useful for the clearance of hypnozoites. Indeed, some of the most advanced anti-malaria vaccines are based on liver stage parasite proteins [22,39,40], including RTS,S/AS01, the only licensed vaccine so far [22]. However, studies of the liver stage of human malaria have lagged behind due to the lack of a physiological in vitro system. Stem cell technology has provided novel avenues to address this stage. In an early study, human iPS cells were differentiated into hepatocyte-like cells in vitro and *Plasmodium* development was successfully established. Effective infection of iPS-derived hepatocytes and maturation of the parasites was demonstrated for murine *Plasmodium* species *P. berghei* and *P. yoelii* and also for the human species *P. falciparum* and *P. vivax* [41]. Cultures could be maintained for up to 55 days. Interestingly, examining the capacity to support infection by *P. falciparum* throughout the differentiation process showed small levels of infection already at the endoderm and hepatic-specified endoderm stages, which increased steadily through hepatoblasts to differentiated hepatocyte-like cells, suggesting that perhaps more immature hepatocytes are also susceptible to the parasite. Testing anti-malarial drugs in this system showed that readily active ones such as atovaquone blocked parasite growth while others that require bioactivation such as primaquine had no effect. This confirms that *in vitro*-generated hepatocytes have lower expression of hepatic metabolic proteins and highlights differences with primary hepatocytes. This limitation was overcome by pre-treatment with small molecules capable of promoting transcriptional regulation of hepatic enzymes, which showed effective inhibition of parasite infection by primaquine. Chemically differentiated mESCs [42] were also shown to support liver stage infections of murine *Plasmodium* species. This approach produces terminally differentiated epithelial-like cells in three days (compared to the 25–35 day hepatic differentiation protocol), which makes it particularly practical for large-scale drug and knock out screening studies. RNA-seq comparisons showed that though chemically differentiated mESCs lack expression of typical hepatocyte markers, they do upregulate host factors previously implicated in malaria infection such as CD18, PKCzeta and hepatocyte growth factor receptor, which make these cells conducive to malaria infection.

### 2.2. Chagas Disease

The American trypanosomiasis, Chagas disease, is caused by *Trypanosoma cruzi*. It is estimated that seven million people are infected worldwide and the mortality rates surpass 10,000 deaths per year [43]. *T. cruzi* is transmitted by triatomine vectors that release trypomastigotes in their faeces during blood meals. The trypomastigotes penetrate through the wound and invade neighbouring cells where they transform into intracellular amastigotes. This proliferative form differentiates into trypomastigotes which are then released into the bloodstream to reach a variety of tissues to initiate the infective cycle. The disease is characterised by an acute phase lasting 1–2 months after infection followed by a reduction in parasitaemia and settling of the parasite in different tissues, generally the heart, stomach, colon and gut [44,45]. The chronic phase of Chagas disease starts without evident clinical manifestations either remaining stable in time or progressing to organ damage, mainly cardiomyopathy or megaviscera in as many as a third of infected people. The progress of Chagas disease is unpredictable and the development of clinical symptoms depends on a wide variety of factors from both the host and the parasite [46]. Development of effective treatment and diagnosis is difficult and hampered by the lack of tools particularly to examine the host factors involved in the disease. While in vitro models have been used in the past, these mainly focused on aspects of parasite biology [47], mostly relying on cell lines that do not recapitulate the features of the cell types most commonly invaded by this parasite. The host–parasite interaction was mainly addressed with animal models, that are not necessarily transferable to the human disease [48]. For example, the most common animal model used is the mouse but the mouse heart rate is 6–10 fold faster than the human heart and the properties of the caridomyocytes are strikingly different, including responsiveness and sensitivity to certain drugs.

Cardiomyocytes (CM) were successfully derived from hiPSCs and shown to be a reliable model for the study of *T. cruzi* infection and assessment of cardiac alterations due to the infection [49]. Drug assays using known and novel compounds showed a similar response between the hiPS-derived CM and the traditional cellular system using Vero cells also detecting some subtle differences [50] that might reflect a more physiological context. One of these, vismione B, showed superior inhibiting capacity than the commonly used benznidazole, particularly when applied after infection without detectable toxicity to the hiPS-derived CM [51]. The full advantage of the iPS system was demonstrated by the derivation of iPSCs from three healthy individuals that were differentiated into CMs. The iPS-derived CMs displayed contractility and expressed specific markers (alpha-actin, troponinT), as well as receptor proteins for *T. cruzi* invasion (bradykinin 1 and 2) [52]. The hiPS-MCs supported *T. cruzi* infection and transformation into all stages, including intracellular amastigotes. Following the kinetics of the infection showed an exponential increase in amastigotes and the release of trypomastigotes in 96 h. Throughout the infection, CM viability decreased by 90% compared to 11% of non-infected CMs. A change in the morphology of the IPS-CMs to an increasingly vacuolated cytoplasm was observed, with significant levels of reactive oxygen and nitrogen species accumulation suggesting infection leads to oxidative stress and apoptosis. Infected iPS-CMs also showed an increase in beat rate and peak height and downregulation of genes involved in Calcium regulation, reproducing the arrhythmic stress and tachycardia characteristic of *T. cruzi* infection. Structurally, infected CMs showed increasing sarcomeric disarray, with loss of troponin T periodicity and cell–cell junctions, as evidenced by changes in connexin 43. Transcriptome comparisons evidenced significant upregulation of 11,000 genes, mainly associated with inflammation, extracellular matrix and metalloproteinases, while sarcomeric and cytoskeletal proteins were downregulated, which correlates with the vacuolar morphology observed. This study demonstrates the feasibility to recapitulate Chagas disease in iPS-CMs as a model and identifies a list of genes that might be important for the pathology and, therefore, represent potential targets for pharmacological intervention and vaccine development.

### 2.3. Leishmaniasis

Over twenty morphologically indistinguishable species of *Leishmania* cause this disease that is transmitted by the bite of infected female phlebotomine sand flies [53]. The promastigotes are injected into the host by the fly during blood meals and are engulfed by phagocytic cells, mainly macrophages. In the host cells, the promastigotes transform into amastigotes which multiply and infect other mononuclear phagocytic cells. Leishmaniasis adopts different forms, ranging from asymptomatic to the visceral form (VL) in which several organs are infected, usually spleen, liver and bone marrow, and can be fatal. Cutaneous leishmaniasis (CL) is characterised by skin sores that can be disfiguring and the less common mucosal leishmaniasis (ML) can arise as a consequence of cutaneous infection with some species. Leishmaniasis is endemic in 98 countries and is estimated to affect over 350 million people, with the incidence of the most common visceral and cutaneous forms varying between 1 and 1.5 million people [54]. Treatment options for this disease are limited and the available drugs are highly toxic, which together with domestic dogs being the main reservoir of the parasites, make the need for chemotherapy for both animals and humans an urgent matter [55,56].

*Leishmania* parasites have been cultured using fibroblast cell lines [57] as well as primary macrophages and monocytes and also established cell lines such as the J774 macrophages [58] and the monocytic leukaemia THP1 [59]. Though these systems are practical for high throughput screens and parasite studies, they have many disadvantages. Primary cells have a limited life span in vitro and immortalised cell lines often carry chromosomal alterations and do not necessarily behave as the original cell type. An early study used adipose tissue-derived mesenchymal stem cells to demonstrate that stem cells can be infected with different species of *Leishmania*: *L. donovani*, *L. major*, *L. tropica* and *L. infantum*. Though the objective of this study was to highlight the necessity for screening for this parasite before stem cell transplantation in endemic countries, it showed that stem cells can be a versatile model to study this parasite [60]. In a recent study, hiPS-derived macrophages were shown to be much better at supporting infection by *L. major* and *L. amazonensis* and very good for *L. mexicana*, as compared with THP1 and primary macrophages [53]. Several compounds were tested in this system with a similar outcome, though IPS-derived macrophages displayed the highest effect in some cases. Despite the higher expense of hiPS-derived macrophages, the fact that they can be stored, preserving their genotypic characteristics, have an unlimited expansion potential while providing higher homogeneity between macrophages from cell lines and donors, represents a major advantage for future studies.

## 3. Simulation of Tissue Environment: Organoids

The traditional cell culture consists of either single cells in suspension or attached to the bottom of culture vessels in monolayers. This system has proven extremely useful for many applications; however, it does not provide information about niches created by different cell types growing in a tissue, which are often environments exploited by a variety of parasites. Stem cell technology offers the possibility to re-create multicellular structures that mimic specific organs. As stem cells are pluripotent, they can be induced to generate the diverse cell types of an organ that assemble into complex, three-dimensional clusters and organise themselves into the architecture and functionality of the desired organ [61,62]. Organoids can be generated from ESCs, iPSCs as well as adult stem cells from specific tissues, supplemented in vitro with a 3D extracellular matrix and developmental factors [63]. Just as the generation of specific cell types from stem cells, organoids are established through specific protocols to first specify the germ-layer (endoderm, mesoderm or endoderm) from which the organ or cell type normally originates, followed by organ-specific induction and maturation (Figure 2). Many types of organoids have been successfully made in vitro including brain, kidney, heart, liver, gut and lung (for review and specific references: [64]) that are of major interest for protozoan infectious disease studies.

### 3.1. Malaria

Cerebral malaria is the most severe clinical manifestation of infection by *P. falciparum*, with a mortality rate of 20–30% and life-changing sequelae for the survivors. The only approaches to gain an understanding of the mechanisms of cerebral malaria have been the *P. berghei* murine model, statistical associations from GWAS analyses and invasive procedures of cerebrospinal fluid analysis. A recent strategy [65] used a hiPS cell line reprogrammed from CD34+ umbilical cord blood cells to develop brain cortical organoids as a model to directly assess the effects of malaria infection by-products and potential treatments. *Plasmodium* parasites induce haemolysis and liberation of heme has been highlighted as a potential cause of blood–brain barrier dysfunction and brain damage. The effects of heme on hiPS-derived organoids recapitulated the inflammatory processes detected in cerebral malaria leading to apoptosis. The attenuating capacity of Neuregulin-1 was confirmed, proposing this system as a viable model for brain injury associated with cerebral malaria. Furthermore, these organoids could be invaluable to test therapeutic agents and to evaluate human genetic variation associated with severe malaria.

### 3.2. Toxoplasmosis

The parasite responsible for this disease is the apicomplexan *Toxoplasma gondii* that is widely distributed throughout the world, with an overall estimate of 30–50% of the world population being infected [66,67]. Infection occurs most often by consumption of undercooked meat of contaminated animals harbouring cysts or food or water contaminated with oocysts shed by infected cats. In the host’s stomach, the oocyst is degraded liberating sporozoites that invade intestinal cells and differentiate into tachyzoites. These are motile, rapidly proliferating forms of the parasite, that transform into bradyzoites, semidormant, slowly dividing stages that form cysts in various tissues. Though cysts can form in virtually any tissue, they predominantly persist in the brain, the eyes and striated muscle, including the heart [68]. A variety of cell systems, mainly transformed human lines such as CHO, HeLa, LM, MDBK, Vero, 3T3, etc; have been used of maintain tachyzoites in vitro and the main primary cells to support culture are human foreskin fibroblasts [69]. These studies have led to an in-depth insight into many processes of the parasite but these systems do not reveal the host–parasite interactions and as a consequence little is known about how epithelial cells respond to the parasite and how the parasite behaves in the infected tissues.

As the first point of interaction with the host, the intestinal epithelium is a paramount tissue to understand the host interaction with *T. gondii* as well as an attractive source of targets for prevention and treatment. This environment has been successfully reproduced in vitro with enteroids, 3D structures mimicking the intestinal architecture derived from primary stem cells and Lgr5+ cells from the base of small intestinal epithelium crypts [70]. Multiple animal sources have been used to obtain these cells and derive enteroids in which infection by the parasite has been achieved [71]. Incorporation of viable, motile parasites into the enteroid lumen was achieved by fragmentation or micro-injection and replicating parasites were observed in epithelial cells [72]. Fragmentation showed invasion of both Paneth and Goblet cells in agreement with previous studies [66,73], while micro-injection had a lower rate of successful invasion, suggesting a different interaction with the parasite or an adverse effect of the procedure. Enteroid-derived epithelial sheets to expose an accessible luminal surface and appropriately polarised epithelial cells to the parasite also supported *T. gondii* infection, and using strains of different virulence, this system showed differential modulation of host proteins in response to either strain. Commonly upregulated proteins included apolipoprotein 1 and 4, chitinase-like protein 4 and pleiotropic regulator 1. Others, such as mevalonate kinase, a key enzyme in sterol and isoprenol synthesis, were upregulated only upon infection with the virulent strain, while the non-virulent one increased serine palmitoyltransferase SPTLC1, which negatively regulates cholesterol efflux. As *T. gondii* lacks sterol and isoprenoid synthesis via the mevalonate pathway, it scavenges cholesterol from the host cells. The importance of mevalonate kinase for parasite survival was confirmed by treatment with the reductase inhibitor, atorvastatin, which caused a marked inhibition of parasite replication. These results show how this novel culture system can contribute to the identification and validation of therapeutic targets.

Chronic Toxoplasmosis includes the formation of cysts within neurones in the central nervous system, which can lead to neuropsychiatric disorders. A first approach to investigate this aspect of the disease involved the development of a culture model using hiPS cells reprogrammed from CD34+ cord blood cells [74]. This cell line was differentiated into neuronal cells as confirmed by expression of neuronal markers, which were efficiently infected by tachyzoites and supported their replication and development of the cyst stage. A more in-depth understanding of the most inaccessible site of *T. gondii* infection was achieved by generating brain organoids from the hESC line H9 [75]. The correct structure and cell types were verified by staining for specific markers, identifying neurones, glia, astrocytes, oligodendrocytes by day 55 of differentiation. These structures can be infected with tachyzoites which proceed through their life cycle transforming into bradyzoites and replicating in parasitophorous vacuoles with cyst formation. Co-staining with cell type markers revealed that *T. gondii* invaded neurones astrocytes and oligodendrocytes with no parasite signal in radial glial cells. By 5 days post-infection, the formation of cysts was evident and the parasites produced were capable of inducing infection in mice. The transcriptome of both, parasite and host cells, revealed a pattern related to life cycle progression in the former and activation of the type I interferon immune response in the latter, demonstrating cerebral organoids as a physiologically relevant model system for this disease and showing that infection of the brain does not seem to be strain-specific. Thus, organoids can be used to study complex parasites in the different environments of the human body, allowing to investigate all aspects of the disease within the same genomic context. This represents a major advantage that has great potential to advance future research.

### 3.3. Criptosporidiosis

The alveolate parasite *Cryptosporidium spp*. from the apicomplexan genus includes many species that infect animals causing this disease. Two main species, *Cryptosporidium hominis* and *Cryptosporidium parvum* infect humans causing respiratory and gastrointestinal disease, principally manifested in diarrhoea with or without a persistent cough. The protective outer shell of these parasites allows survival in open environments and provides tolerance to chlorine; therefore, a major route of contamination is drinking and recreational water. In healthy individuals, infection symptoms are generally mild but they can become severe and even fatal in immunodeficient, malnourished and very young individuals [76]. Unsurprisingly, the majority of cryptosporidium-related deaths occur in low income or unstable countries, affecting mainly children under five years of age with the 2016 statistics reporting 48,000 deaths mostly in sub-Saharan Africa [77]. Even mild or asymptomatic infections can have long-term consequences with significant impact in adult life. Problems associated with this disease include under-diagnosis, very limited treatment options and crucially the lack of a vaccine [78]. All of these interventions need an in-depth understanding of the pathophysiology of *Cryptosporidium* but these types of studies have been hindered by the lack of reliable in vitro culture systems.

Several approaches have been used for *Cryptosporidium* culture [79], based on two-dimensional cultures using human colon cancer cell lines such as HCT-8 and Caco-2 or monolayered cultures of primary cells [80,81], as well as 3D cultures with these lines alone or in co-culture with primary cells that can form tridimensional structures in vitro [82,83]. None of these systems can fully reproduce the host–parasite interactions that occur in vivo and have limitations in the scale and viability of the oocysts produced. 3D culture models derived from mouse explants have also been developed that support culture for longer periods but are limited by the lifespan of the tissue explants [84].

Recently, intestinal and lung organoids or enteroids derived from tissue biopsies obtained from healthy human donors have been developed that represent a physiological in vitro model supportive of the *Cryptosporidium* life cycle [85,86]. The small intestinal organoids were established from duodenal biopsies, from which small intestinal crypts were isolated and cultured in the presence of cytokines and factors. The lung tissue was dissociated and cultured in a specified medium. The organoids were microinjected with the parasite, which infected, replicated and completed its entire life cycle, demonstrating that these organoids are a physiologically relevant model. The development of lung organoids that support *Cryptosporidium* life cycle offers a great opportunity to study infection of the respiratory tract, which though it has been reported remains largely unknown. Transcriptomic analysis of the differentiated organoids after infection identified early changes involving genes related to cytoskeleton and cell mobility in the lung system, suggesting structural changes induced by the parasite in the host cells. Later time points showed a dramatic increase in the expression of genes associated with the type I interferon pathway. The response of intestinal enteroids to the parasite involved fewer genes, but the upregulation of type I interferon pathway genes was also observed, pinpointing this pathway as a major anti-parasite effector. The use of both types or organoids revealed differences in the response of different tissues to the parasite that might become important for the development of novel treatment strategies. The transcriptome of *C. parvum* revealed substantial modulation of gene expression in both types of organoids with genes related to growth and metabolic activity at the early time point reflecting the high proliferation observed. The later time point (72 h) was characterised by an increase of oocyst genes, confirming the progression of the parasite’s life cycle.

Murine enteroids have also been used to understand the impact of the parasite on the intestinal environment [87], showing that infection leads to a decrease in proliferation and expression of intestinal stem cell markers Lgr5 and Sox9. This suggested that the parasite could inhibit intestinal stem cell (ISC) function through attenuation of the Wnt/Beta-cat signalling pathway resulting in apoptosis and senescence. A novel murine model system was developed from ISCs grown as spheroids creating an air–liquid interface (ALI) that improved culture conditions for longer-term *C. parvum* maintenance [88]. This system supports the full life cycle, parasite expansion and generation of infective oocysts transmissible in vitro and in mice. Genetically modified parasites were used to demonstrate successful genetic cross in vitro and validating this system as a viable novel tool for the study of *Cryptosporidium* [89]. Four types of compounds targeting different stages of the parasite’s life cycle were tested for anti-*C. parvum* activity using this system. Comparison with standard culture in the HCT-8 colon cancer cell line showed some differences in potency and importantly, that the more physiological ALI system makes it possible to assess the efficacy of the compounds on different stages as well as the concentration and timing to prevent parasite recovery after treatment [90]. This study highlights the possibility of measuring the impact compounds and drugs have on the host cells and, therefore, has the potential to predict those that might be amenable for clinical use.

## 4. Concluding Remarks and Perspectives

Protozoan parasites represent an enormous burden for public health worldwide, not just in the cost of lives and physical impairment but with grave consequences on productivity, economic activity and development. Therefore, substantial efforts have been invested in finding novel approaches for the prevention and treatment of these diseases. However, the complexity of these parasites, with life cycles involving vectors, natural reservoirs and stages in a variety of tissues in the human body, make the search for therapeutic tools very difficult. One of the main issues has been the lack of practical and adequate systems to maintain these parasites in the laboratory, also allowing to assess the fundamental role of the host in determining the onset and progress of the infection. The realisation of the host’s participation in these diseases has renewed interest in searching better, more physiologically relevant, easily manipulatable and scalable systems to culture and study these parasites.

The advances in stem cell technology have opened new avenues to develop in vitro cellular systems [63,64] that can be adapted to provide more physiological environments to grow protozoan parasites. A variety of stem cell lines (ESC or iPS) have been established that can be expanded, cultured, stored, genetically edited and differentiated into any desired cell type, providing unlimited access to cell types as well as mini-organs for modelling infectious diseases. These resources have already been proven to support several protozoan parasites and to recapitulate the natural infection, demonstrating their potential to be expanded to other parasites that have eluded in vitro maintenance so far, such as *P. vivax* or *C. hominis*. Even parasites with well-established culture methods such as *Giardia duodenalis* or *Entamoeba spp*. could benefit from organoid systems to investigate the events in the host that lead to disease and pathology. Furthermore, the ability to reprogramme terminally differentiated cells from epithelium or blood into iPS lines avoids ethical issues stemming from the use of ESCs and adds the possibility of perpetuating in the laboratory pluripotent lines from a variety of genetic backgrounds. This offers the opportunity to study rare genotypes, confirm GWAS data and derive lines from patients with known clinical histories. The versatility of this system means that different cell types of the same genetic background can be generated to examine different stages of the parasites and the diseases. Additionally, methods for isolating adult stem cells from various tissues have also been developed that complement the set of tools for the understanding of human biology, and the host factors involved in the interaction with protozoan parasites.

In these systems, parasites can be studied in an approximation to their natural environment, which is starting to provide important clues about drug intake, metabolism and dose efficacy. Assessment of the exact life cycle stage affected by different drugs, invasion of different tissues and behaviour of the parasites in these particular environments will lead to the identification of vital proteins essential for parasite invasion and survival. It is precisely these crucial proteins that represent ideal targets for vaccine and drug development.

As any in vitro system, stem cell models also have some limitations that need to be taken into consideration. As these are pluripotent cells, they can differentiate spontaneously; therefore, strict care needs to be invested in their maintenance, culture and storage. Differentiation protocols have been developed for many cell types, but they can result in heterogeneous populations of cells, which need to be characterised according to surface markers and sometimes sorted by various methods. Another common problem is that in vitro generated cell types correspond to the more foetal counterparts rather than adult cells, which can affect parasite invasion or growth, or the processing of particular drugs. The efficiency of the differentiation process can be greatly improved by the protocol used and also by genetic engineering, providing the crucial transcription factors to guide the cells towards the desired paths [91]. Though it is expensive to derive, maintain and differentiate stem cells, these approaches are constantly improving and becoming more accessible. Stem cell technologies are in line with the 3Rs (Replacement, Reduction and Refinement) principles avoiding and reducing costly animal experimentation, at the same time providing stable and more physiological models for the study of infectious diseases. With the increasing use and adaptation to study protozoan parasites, both fields will be able to nurture each other for the continuous development of novel approaches that will open new horizons for the understanding of these organisms and find new ways to prevent and treat these diseases.

## Figures and Tables

**Figure 1 vaccines-09-00105-f001:**
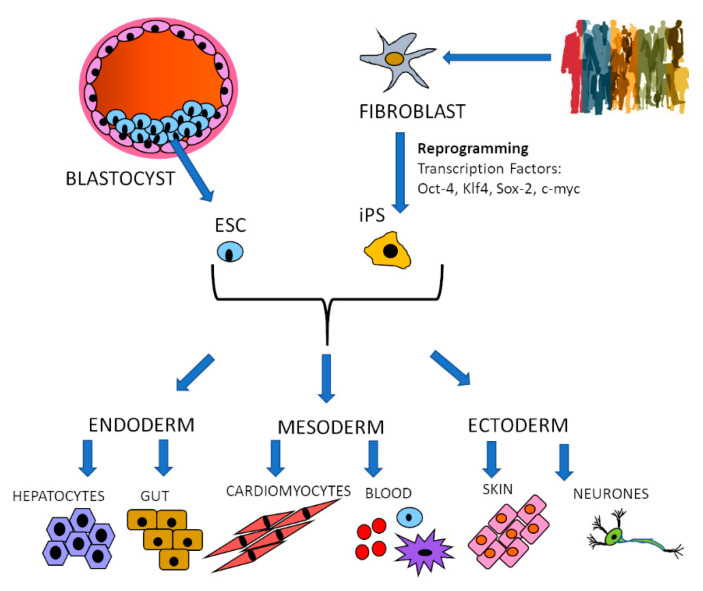
Stem cell differentiation. Embryonic stem cells (ESC) are obtained from the inner mass of the blastocyst, while induced pluripotent stem cells (iPS) are reprogrammed from terminally differentiated cells such as fibroblasts by transduction with transcription factors. Stable cell lines are established that can be differentiated into the three germ layers: endoderm, mesoderm and ectoderm and into the cell types normally originating from each of them. The differentiation process is guided by a defined medium that provides cues for the cells to follow specific developmental paths.

**Figure 2 vaccines-09-00105-f002:**
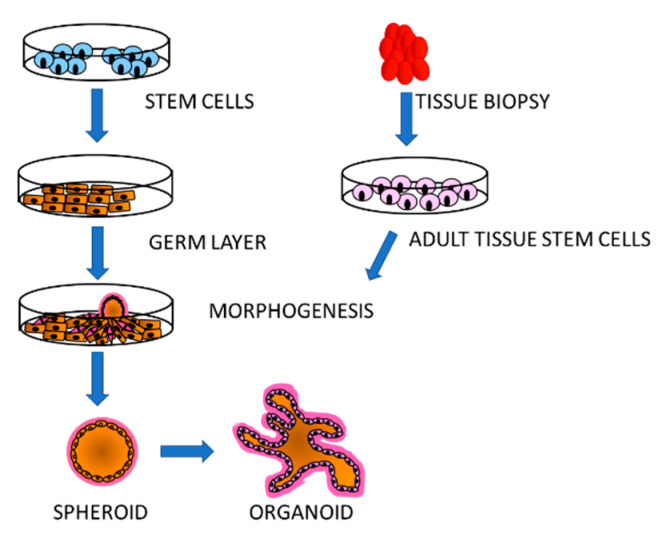
Organoid derivation. These tridimensional structures resembling mini-organs can be generated from any type of stem cell: ESC, iPS, adult stem cells from established lines or biopsies. The cells are differentiated into the desired germ layer: endoderm, mesoderm or ectoderm, from which the stem cells differentiate into the different cell types constituting the organ of interest. The cells assemble into spheroids that then develop and grow into organoids. The differentiation and assembly processes are guided by defined medium and supporting substrate that promote differentiation and formation of specific organ types.

## Data Availability

No data has been generated related to this manuscript.

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
