# Peer review of "The Stem Cell Revolution Revealing Protozoan Parasites’ Secrets and Paving the Way towards Vaccine Development"

_vaccines, 2021, doi:10.3390/vaccines9020105_

Round 1

Reviewer 1 Report

The review provides good insights and possibilities on vaccine development but limited to some protozoans. Conclusions are quite long; can be shorten.

Author Response

Many thanks for the reviewer’s comments. The manuscript indeed concentrates on protozoan parasites that infect humans and for which stem cell technologies have been used in order to have some data to discuss. By raising awareness of these applications, this manuscript hopes to stimulate a wider use of these tools to study other parasites and diseases taking full advantage of the technology and advancing the field. This point has been made in the conclusions section.

The conclusions section has been shortened.

Reviewer 2 Report

This review provides a nice update on the use of stem cell derived organ-specific cells for studying parasite development and host-parasite interaction. There is no double that specialized cells derived from stem cells provide important platforms for these studies. However the relatively high cost and complicated procedures for cell culture may prevent the wide use of the techniques. Figure 1 provides an nice overview of the procedures. It would be better to provide additional details including addition of cytokines for derivation of a specific cell type (such as red blood cells). The uses of cells derived from individual human hosts to examine the effects human genetic variation on infectious organisms and to confirm correlations from epidemiologic and Genome-Wide Association Studies (GWAS) are good ideas if a relatively large number of cell lines can be obtained from individuals with reasonable cost and time.

The title suggests that the article is about revealing parasite secrets and vaccine development; however, the examples cited are mostly host responses to parasite infections, parasite differentiation, or host-parasite interactions. There is no discussion on how the techniques can be used for vaccine development. A paragraph to connect the title and vaccine development or a change of the title is necessary.

Minor issues:

1, line 62: a space with ‘systemand’.

2, Line 62, “animal models cannot give an insight into the impact”: Better to replace ‘cannot’ with ‘may not’. Animal models can provide important insight.

3, Line 90, ‘Embryonic Stem Cells (ESC)’: ESC should be abbreviated the first time used above.

4, Line 132, ‘while  the whole life cycle can only be studied in the mouse model”: Life cycle can be studied in non-human primates and humanized mice now.

5, Line 139-141: It would be helpful to provide more details on how HSCs were used in the study, and why other cell line could not be or were not used?

6, Line 141, P. falciparum: italic.

7, Line 142, GYPA, BSG: spell out.

8, Line 154, ‘Immortalised HSCs were generated from peripheral blood were able to support invasion by P. falciparum and also P. vivax’. Please cite the reference.

9, Line 167, ‘ie infection’?

10, Line 170, ‘including the only 170 licenced vaccine so far’: Please name the vaccine and cite reference.

11, References: there are many references missing page numbers, and some not aligned properly.

Author Response

Comments and Suggestions for Authors

Many thanks for the reviewer's insightful comments, please find specific answers below:

This review provides a nice update on the use of stem cell derived organ-specific cells for studying parasite development and host-parasite interaction. There is no double that specialized cells derived from stem cells provide important platforms for these studies. However the relatively high cost and complicated procedures for cell culture may prevent the wide use of the techniques. Figure 1 provides an nice overview of the procedures. It would be better to provide additional details including addition of cytokines for derivation of a specific cell type (such as red blood cells). The uses of cells derived from individual human hosts to examine the effects human genetic variation on infectious organisms and to confirm correlations from epidemiologic and Genome-Wide Association Studies (GWAS) are good ideas if a relatively large number of cell lines can be obtained from individuals with reasonable cost and time.

While it is true that derivation of IPS cells and differentiation protocols for different cell types is laborious and expensive, animal models are also expensive, many times do not reflect the human scenario and present ethical issues regarding the use of animals for research. The techniques and tools for reprogramming, editing and differentiation are constantly improving and new protocols and strategies are being developed that make the experimental processes easier. Analysis of the significance of genomic variation in this way indeed requires several cell lines though not necessarily high numbers of them, but it is the only way to assess its contribution to parasitic infection and validating the GWAS findings. It also has the advantage that different cell types can be generated from the same IPS line, therefore having the same genetic background. This is a unique opportunity for the study of parasites that infect several tissues such as malaria or leishmania.

These aspects of stem cell technologies have been made clearer in the conclusions section.

The title suggests that the article is about revealing parasite secrets and vaccine development; however, the examples cited are mostly host responses to parasite infections, parasite differentiation, or host-parasite interactions. There is no discussion on how the techniques can be used for vaccine development. A paragraph to connect the title and vaccine development or a change of the title is necessary.

Stem cell technologies have only recently started to be applied to the study of parasitic infections and as such, most of the work done so far has aimed at demonstrating the feasibility of these approaches. The fact that parasite infection and development can be replicated in these systems, opens the way for further and more detailed studies. Nevertheless, even the data we have now already identifies novel candidates for vaccine and drug development and provides new insights into parasite biology revealing new aspects of life cycles and interactions with the host that could be exploited for drug and vaccine development in the future.

This has been made clearer in the text (end of the introduction and conclusion section), to make a stronger link with the title.  

Minor issues:

1, line 62: a space with ‘systemand’.

Corrected

2, Line 62, “animal models cannot give an insight into the impact”: Better to replace ‘cannot’ with ‘may not’. Animal models can provide important insight.

This change has been made.

3, Line 90, ‘Embryonic Stem Cells (ESC)’: ESC should be abbreviated the first time used above.

Corrected

4, Line 132, ‘while  the whole life cycle can only be studied in the mouse model”: Life cycle can be studied in non-human primates and humanized mice now.

This has been added:

While the whole life cycle can be studied using mouse models and more recently humanised mice (32739836, 30631319) and non-human primates (27748213) have been developed for human malaria parasites.

5, Line 139-141: It would be helpful to provide more details on how HSCs were used in the study, and why other cell line could not be or were not used?

This section was improved to:

An initial effort to manipulate gene expression in the anuclear erythrocytes used HSCs and RNAi-based knock down strategies to screen for host determinants of malaria infection. An shRNA library targeting the erythrocyte-specific blood groups was transduced into CD34+ HSC, which were then induced to proliferate and differentiate towards the erythroblast stage. Infection with fluorescent P. falciparum parasites and quantification of each shRNA abundance in the infected population of cells identified CD55 as an important host factor for P. falciparum invasion involved in parasite attachment to the erythrocyte.

6, Line 141, P. falciparum: italic.

Corrected

7, Line 142, GYPA, BSG: spell out.

Corrected

8, Line 154, ‘Immortalised HSCs were generated from peripheral blood were able to support invasion by P. falciparum and also P. vivax’. Please cite the reference.

This is one piece of work, part of reference 37. It has been added again to make this clear.

9, Line 167, ‘ie infection’?

The abbreviation of id est has been corrected.

10, Line 170, ‘including the only 170 licenced vaccine so far’: Please name the vaccine and cite reference.

The vaccine is RTS,S/AS01 and the reference is 22, this has been added

11, References: there are many references missing page numbers, and some not aligned properly.

Corrected

Reviewer 3 Report

Overall:

  • Needs a lot of copy-editing (many grammatical and spelling mistakes).
  • Figure 2 seems a little simple. Perhaps combine Fig. 1 and Fig. 2 into one larger figure?
  • General numbering system is off. The subsections under “SPECIFIC CELL TYPE GENERATION: STEM CELLS” should be 2.1, 2.2, etc. and the section “SIMULATION OF TISSUE ENVIRONMENT: ORGANOIDS” should be section 3 with subsections 3.1, 3.2, etc.

Specific comments:

  • Line 135: I’m assuming “assess to the host cells” is referring to ability to genetically manipulate the host cells, not physical assess to the cells. Needs to be clearer.
  • Line 144: Include the abbreviation (BSG) after the work “Basigin” so it is not confusing on line 149 when it is referred to as BSG.
  • Section on Chagas Disease (starting on line 197): There is no general summary on the life cycle of trypanosomes like in other sections so that when the reader gets to lines 230-231, it is unclear what amastigotes and promastigotes are to those without a parasitology background.
  • Line 350: Need citations after “with previous studies”.
  • Line 389: Should be “Cryptosporidiosis” not “Criptosporidiasis”
  • Lines 391-392: hominis and C. parvum are the two main species that infect humans, but they are not the only two.
  • Line 449: The cited reference does not indicate a “significant” reduction in potency of compounds in ALI vs HCT-8 cells, other than for nitazoxanide. Most compounds have very comparable EC50s, and this statement should be corrected.

Author Response

Comments and Suggestions for Authors

Many thanks to the reviewer for their insightful comments, please find specific answers below:

Overall:

  • Needs a lot of copy-editing (many grammatical and spelling mistakes).

The whole manuscript has been re-checked for grammar and spelling (Note: British English spelling is used throughout the manuscript).

  • Figure 2 seems a little simple. Perhaps combine Fig. 1 and Fig. 2 into one larger figure?

An overall figure combining all the stem cell technologies was originally considered. However, when designing such a figure, it was realised that it would be far too big and consequently some of the detail would be inevitably lost. Separating the cellular and the organoid approaches seems also more fitting with the flow of the text and therefore deemed a better option. Figure 2 was improved by adding the derivation of organoids from tissue to reflect the technology available which adds a bit of complexity.

  • General numbering system is off. The subsections under “SPECIFIC CELL TYPE GENERATION: STEM CELLS” should be 2.1, 2.2, etc. and the section “SIMULATION OF TISSUE ENVIRONMENT: ORGANOIDS” should be section 3 with subsections 3.1, 3.2, etc.

This has been corrected 

Specific comments:

  • Line 135: I’m assuming “assess to the host cells” is referring to ability to genetically manipulate the host cells, not physical assess to the cells. Needs to be clearer.

Access has been changed to: editing of the host erythrocytes

  • Line 144: Include the abbreviation (BSG) after the work “Basigin” so it is not confusing on line 149 when it is referred to as BSG.

Corrected

  • Section on Chagas Disease (starting on line 197): There is no general summary on the life cycle of trypanosomes like in other sections so that when the reader gets to lines 230-231, it is unclear what amastigotes and promastigotes are to those without a parasitology background.

The following has been added at the start of the Chagas disease section:

  1. cruzi is transmitted by triatomine vectors that release trypomastigotes in their faeces during blood meals. The trypomastigotes penetrate through the wound and invade neighbouring cells where they transform into intracellular amastigotes. This proliferative form differentiates into trypomastigotes that are released into the bloodstream to reach a variety of tissues to initiate the infective cycle.
  • Line 350: Need citations after “with previous studies”.

These were added

  • Line 389: Should be “Cryptosporidiosis” not “Criptosporidiasis”

Corrected

  • Lines 391-392: hominis and C. parvum are the two main species that infect humans, but they are not the only two.

This has been made clear: Two main species…

  • Line 449: The cited reference does not indicate a “significant” reduction in potency of compounds in ALI vs HCT-8 cells, other than for nitazoxanide. Most compounds have very comparable EC50s, and this statement should be corrected.

Showed some differences in potency and importantly, that the more physiological ALI system makes it possible to assess efficacy of the compounds for different stages as well as the concentration and timing to prevent parasite recovery after treatment.